# "What can her body do?" Reducing weight stigma by appreciating another person's body functionality

Jessica M. Alleva[1]*, Kai Karos[1,2], Angela Meadows[3], Moon I. Waldén[1], Sarah E. Stutterheim[4], Francesca Lissandrello[5], Melissa J. Atkinson[6]

1 Department of Clinical Psychological Science, Maastricht University, Maastricht, Netherlands, 2 Centre for the Psychology of Learning and Experimental Psychopathology, KU Leuven, Maastricht, Netherlands, 3 Department of Psychology, Western University, London, Canada, 4 Department of Work and Social Psychology, Maastricht University, Maastricht, Netherlands, 5 Sigmund Freud University, Milano, Italy, 6 Department of Psychology, University of Bath, Bath, England

* jessica.alleva@maastrichtuniversity.nl

## Abstract

### Objective

Weight stigma is prevalent across multiple life domains, and negatively affects both psychological and physical health. Yet, research into weight stigma reduction techniques is limited, and rarely results in reduced antipathy toward higher-weight individuals. The current pre-registered study investigated a novel weight stigma reduction intervention. We tested whether a writing exercise focusing on *body functionality* (i.e., everything the body can do, rather than how it looks) of another person leads to reductions in weight stigma.

### Method

Participants were 98 women ($M_{age}$ = 23.17, Range = 16–63) who viewed a photograph of a higher-weight woman, "Anne," and were randomised to complete a writing exercise either describing what "Anne's" body could do (experimental group) or describing her home (active control group). Facets of weight stigma were assessed at pretest and posttest.

### Results

At posttest, the experimental group evidenced higher fat acceptance and social closeness to "Anne" compared with the active control group. However, no group differences were found in attribution complexity, responsibility, and likeability of "Anne".

### Conclusions

A brief body functionality intervention effectively reduced some, but not all, facets of weight stigma in women. This study provides evidence that functionality-focused interventions may hold promise as a means to reduce weight stigma.

**Data Availability Statement:** The data have now been uploaded to a repository, and are accessible here: https://doi.org/10.34894/0KSIPK.

**Funding:** The author(s) received no specific funding for this work.

**Competing interests:** The authors have declared that no competing interests exist.

## Introduction

Stigmatisation is a socially and culturally constituted process whereby a person is identified as different and then devalued, leading to status loss and discrimination [1]. In particular, *weight stigma* is characterised by negative attitudes towards a person based on their body weight and size, and is typically directed toward individuals with higher body mass index (BMI) who are perceived to be "overweight" or "obese" [2]. Weight stigma is prevalent across numerous life domains—including interpersonal relationships, education, employment, housing, health care, and media (for a review, see [3])—and has severe consequences. For instance, experiencing weight stigma has been associated with increased levels of maladaptive eating behaviours, including binge eating, loss-of-control eating, and emotional eating [4], motivation to avoid exercise, and lower levels of moderate and vigorous physical activity [5], as well as high-risk health behaviours [6], allostatic load [7], risk of chronic disease [8], higher rates of depression and anxiety [9,10], substance use disorders [11], suicidal ideation [12], and mortality [13], independent of BMI (for reviews, see [3,4,9,10,14,15]). Further, experimental studies have confirmed a causal link between exposure to weight-stigmatising content or experiences and increased calorie intake [16–18] and increased cortisol reactivity [19,20].

Yet, despite the pervasiveness and negative consequences of weight stigma, few studies have investigated weight stigma reduction techniques, with mixed results (for reviews, see [21,22]). Of the interventions that are founded on a named theoretical approach, most are based on *attribution theory*, which suggests that negative feelings towards higher-weight individuals are due to the targets' perceived culpability for their weight [23,24]. Interventions that aim to alter beliefs about the causes of "obesity" and the supposed controllability of body weight tend to be effective in changing knowledge about the constructs targeted, but they are generally less effective at reducing dislike [21,22]. One possible explanation for this lack of success is that interventions aiming to reduce culpability for a negatively valued attribute (fatness), nevertheless continue to assign negative value to higher-weight bodies [25]. Somewhat greater success has been achieved by interventions aiming to increase empathy toward higher-weight individuals—although effect sizes are still only small-to-moderate [22]. Again, however, significant heterogeneity has been reported across studies, and empathy-building approaches may also produce paradoxical results, increasing anti-fat attitudes in some individuals [26]. Other useful approaches include manipulating apparent social norms around anti-fat attitudes and interventions founded upon a size-acceptance paradigm [22], whereby the sociocultural origins of modern anti-fat attitudes are identified and higher-weight bodies are presented as a natural part of a continuum of body sizes across the population, and not inherently problematic [27,28]. Nevertheless, the literature in this area remains relatively sparse, highly heterogeneous, and often consists of atheoretical, small, non-controlled interventions that preclude inferences about causality [21,22].

Accordingly, the aim of this study was to investigate a novel technique for reducing weight stigma. To do so, we drew from emerging research from the field of *body image*—that is, how individuals think and feel about their body—showing that appreciating the functionality of one's own body is currently the most effective technique to enhance positive body image (for a review, see [29]). *Body functionality* refers to everything that the body is able to *do*, and comprises six domains: (a) internal processes (e.g., digesting food, absorbing vitamins); (b) physical capacities (e.g., walking, stretching); (c) bodily senses and sensations (e.g., seeing, hearing); (d) creative endeavours (e.g., singing, drawing); (e) self-care (e.g., showering, sleeping); and (f) communication with others (e.g., via body language, eye contact; [30]). *Functionality appreciation* refers to appreciating, honouring, and respecting the body for what it is able to do (regardless of whether the body can function "well" in all domains; [31]). In functionality-based body image interventions, participants are typically guided to write about what their body is able to

do, and why those functions are valuable to them [30]. Both single- and multi-session variants of this approach have been shown to improve body image among women and men, with effects lasting up to 1-month follow-up [30,32–38].

The functionality-based approach to improving body image is grounded in *objectification theory* [39]. According to this theory, women are often valued and evaluated based predominantly on their physical appearance; consequently, they may develop the tendency to engage in *self-objectification*, whereby they come to value and evaluate their *own* body based predominantly on its physical appearance, from a third-person observer perspective. Self-objectification contributes to numerous health risks for women, including negative body image and disordered eating (for reviews, see [40,41]). In line with objectification theory, appreciating one's body functionality is proposed to minimise the tendency to emphasise one's physical appearance, thereby improving body image [30]. Indeed, studies have shown that appreciating one's body functionality via structured writing exercises leads to reduced self-objectification and to viewing one's own body in more complex, holistic terms [30,34].

Inspired by this line of research, the present study investigated whether appreciating the body functionality of *another* person—in particular, a woman with high BMI—would lead to reductions in weight stigma. Why might this approach be effective? Research on objectification theory has shown that viewing *other* people predominantly in terms of their physical appearance (i.e., engaging in objectification) is also related to negative consequences, including viewing others as "less fully human" and internalising cultural appearance standards that idealise thinness [42–44]. Weight stigma may involve objectification of others, as the stigmatised person is evaluated predominantly on how they look, based on their body weight. Indeed, evidence suggests that high-BMI individuals are viewed as less human, a finding that holds across the weight spectrum, and that dehumanisation predicts support for policies that discriminate against those with high BMI [45].

In line with objectification theory and the prior research on body functionality described above, when individuals appreciate the body functionality of a person with high BMI, this could encourage them to view that person's body in a more complex, holistic manner, thereby reducing an overemphasis on body weight, and mitigating weight stigma. It is also important to note that many of the negative attitudes towards people with high BMI involve negative assumptions about their body functionality, specifically with respect to the domains of internal processes and physical capacities; for example, that they are "unhealthy," "unfit," and "lazy" [3]. Encouraging individuals to appreciate the diverse domains of body functionality could broaden their conceptualisations of the other person's body functionality and contribute to reductions in weight stigma.

## The present study

The primary research question of this study was: Does appreciating the body functionality of another person lead to reductions in weight stigma? Participants were randomised to complete a writing exercise that elicited either functionality appreciation for a woman with high BMI, or descriptions of her home (as an active control). Facets of weight stigma were assessed at pretest and posttest. It was hypothesised that the experimental group would evidence lower weight stigma at posttest compared with the active control group.

A secondary research question was: Are the effects of the present intervention approach moderated by pre-existing levels of functionality appreciation? Scholars have called for a more tailored approach to interventions, whereby a "match" is made between the change technique and those who will benefit most from it [46]. Individuals who are *higher* in functionality appreciation might respond better to the intervention approach because they are more oriented

toward functionality appreciation compared with those lower in functionality appreciation. Yet, individuals who are *lower* in functionality appreciation might benefit more from the intervention, because they would stand more to gain from contemplating body functionality. As a theoretically sound argument could be made in either direction, a specific hypothesis was not formulated for this research question.

## Methods

### Participants and procedure

A priori calculations indicated that 100 participants were needed for the planned statistical analyses (assuming $\alpha = .05$, 80% power, and a small-to-moderate effect size; cf. [30]). The final sample comprised 98 women (their demographic characteristics are shown in Table 1). An

**Table 1. Participants' demographic characteristics.**

|  | *M (SD)* | *Range* |
|---|---|---|
| **Age** | 23.17 (6.43) | 16–63 |
| **Body mass index** | 21.13 (2.56) | 16.53–31.49 |
|  | *n* | *%* |
| **Self-classified weight** |  |  |
| Very underweight | 1 | 1.0 |
| Moderately underweight | 4 | 4.1 |
| Slightly underweight | 4 | 4.1 |
| Neither underweight nor overweight | 73 | 74.5 |
| Slightly overweight | 11 | 11.2 |
| Moderately overweight | 4 | 4.1 |
| Very overweight | 1 | 1.0 |
| **Do you identify as Fat?** |  |  |
| Yes | 9 | 9.2 |
| No | 89 | 90.8 |
| **Ethnic background** |  |  |
| German | 18 | 18.4 |
| Dutch | 14 | 14.3 |
| Other Western European | 7 | 7.1 |
| Eastern European | 14 | 14.3 |
| Southern European | 9 | 9.2 |
| Northern European | 1 | 1.0 |
| Asian | 10 | 10.2 |
| South American or Latin American | 4 | 4.1 |
| North American | 2 | 2.0 |
| African | 1 | 1.0 |
| Mixed | 17 | 17.4 |
| Rather not say | 1 | 1.0 |
| **Sexual Orientation** |  |  |
| Heterosexual | 86 | 87.8 |
| Bisexual | 5 | 5.1 |
| Lesbian | 1 | 1.0 |
| Pansexual | 2 | 2.0 |
| Queer | 1 | 1.0 |
| Other | 1 | 1.0 |
| Rather not say | 2 | 2.0 |

additional three women had taken part in the study but their data were excluded because they did not comply with the instructions of their assigned writing exercise.

This study was approved by the Ethics Review Committee Psychology and Neuroscience at Maastricht University (ethics approval code OZL_159_15_12_2015_S15), and was preregistered on AsPredicted (protocol #31330, see http://aspredicted.org/blind.php?x=ed2uv6). No changes were made to the study protocol after preregistration. Participants were recruited for a study about "techniques to influence imagination skills" (cf. [33]) via flyers on campus, social media (e.g., accounts of the research group), and the university's online system for participant recruitment.

The pretest took place online. Participants signed an electronic informed consent sheet and completed the Fat Attitudes Assessment Toolkit (FAAT; [47]), Functionality Appreciation Scale (FAS; [31]), demographic items, and filler items. A laboratory session took place approximately 1 week later (± 2 days). First, the experimenter told the participant that she would complete an "imagination exercise." The experimenter presented an A4 framed colour photograph of "Anne," a woman with high BMI (details below), and placed it next to the computer. The participant was given a booklet of instructions and was asked to confirm that she understood them before the experimenter left the room. The participant was instructed to write continuously for 15 min. After the 15min had passed, the experimenter re-entered the lab, collected the photograph of "Anne" and the instruction booklet, and opened the posttest questionnaires. The posttest questionnaires comprised items assessing social closeness and likeability of "Anne," the FAAT, filler items, perceived body weight of "Anne," and an awareness check. Participants received a €7.50 voucher or course credit, and were debriefed via email at study completion.

## Experimental manipulations

Participants were told they would be writing "about the woman in the photograph next to you, named Anne." *The experimental group* was asked to "imagine, and write about, the many things that her body is able to *do*." They read a list of examples of body functions, categorised by domain (e.g., creative endeavours). In their writing, participants were instructed to "take your time, really let go, and imagine the many different things that her body *can do*." They were told they could write about as many functions as they liked, but were encouraged to consider the different domains of body functionality. To underscore appreciation for her body functionality, they were asked to reflect on, "Why are these body functions important to Anne?".

*The active control group* was asked to "imagine, and write about, the *house* in which Anne lives, and the *details* of her house." They read a list of examples of home features, categorised by domain (e.g., interior design). In their writing, participants were instructed to "take your time, really let go, and imagine the many different details of *her house*." They were told they could write about as many features as they liked, but were encouraged to consider the different domains of home features. "Anne's" home was chosen as a focus of this writing exercise, to ensure that participants in the active control group would also write about something personal to "Anne," but that was not directly related to her body.

All participants were told that they would spend 15 min on the writing exercise, and that they should write as much as they could in that time, but not worry about "finishing" it. They were reminded that their writing would be confidential and anonymous, and were told not to worry about spelling, sentence structure, or grammar. The full instructions of the writing exercises are provided in the Supporting Information (S1 File).

## Photograph of "Anne"

The photograph of "Anne" was selected from Obesity Canada's gallery "that portrays individuals with obesity in ways that are positive and non-stereotypical" [48]. Five photographs of different women were selected by the fourth and sixth authors. These photographs were discussed at a meeting with the research group of the first author. Based on their feedback, two of the photographs were chosen for pilot testing; they were evaluated by nine undergraduate women (not involved in the study) from 1 = *very underweight* to 7 = *very overweight*. The photograph with the highest ratings ($M = 6.11$ vs. $M = 5.89$) was chosen. This photograph was a full-body image of a White woman wearing form-fitting jeans and a sweater, standing, smiling, and looking straight into the camera (see Supporting information, S2 File). The name "Anne" was chosen because it is a common name in Dutch, German, and English (the main languages spoken in the region where the study was conducted).

## Measures

**Fat Attitudes Assessment Toolkit (FAAT; [47]).**  The FAAT comprises the following four scales. All items are rated from 1 = *strongly disagree* to 7 = *strongly agree*.

The *Fat Acceptance Scale* contains 32 items that can be divided into five subscales: Empathy (7 items; e.g., "Fat people face discrimination in many areas of life"), Discrimination (9 items; e.g., "Discrimination due to fatness leads to a denial of human rights"), Size Acceptance (6 items; e.g., "We should celebrate all bodies"), Attractiveness (5 items; e.g., "Fat people are sexy") and Health (5 items; e.g., "Fat people are not necessarily unhealthy"). A composite score of the Fat Acceptance Scale items was used (cf. [47]), with higher mean scores indicating higher levels of fat acceptance and more positive evaluations of fat people.

The *Attribution Complexity Scale* (9 items) comprises the General Complexity Subscale (6 items; e.g., "There are many factors that cause people to be fat") and the Socioeconomic Complexity Subscale (3 items; e.g., "There are factors relating to social inequality that cause people to be fat"). A composite score of the Attribution Complexity Scale items was calculated (cf. [47]), with higher mean scores indicating endorsement of multiple external causes for fatness, which can be deemed outside of personal control.

The *Responsibility Scale* (6 items; e.g., "Fat people lack willpower") focuses specifically on individual attribution. Item scores are reverse-coded and then averaged, with higher scores indicating that fewer factors relating to personal responsibility are assigned to the causes of fatness.

The *Body Acceptance Scale* (4 items; e.g., "I feel happy about my weight") assesses participants' appraisal of their own body weight. The data derived from this scale were not analysed, as they were not central to the research questions.

Last, the FAAT comprises one optional item, "Do you identify as Fat?" (*yes/no*). This item was included to further describe the demographic characteristics (see below) of this sample.

Among adults in the U.S., FAAT item scores have demonstrated good test-retest reliability, construct validity, and internal reliability ($\alpha = .84$ to.96; [47]). Further, FAAT scores have not been correlated with socially desirable responding [47]. Cronbach's $\alpha$ for the FAAT (and remaining measures) in this study are reported in Table 2. Note that the FAAT uses the term "fat," in line with modern critical fat discourse (see [47]); this term is used in this manuscript when referring to the outcomes from the FAAT.

**Social closeness to "Anne".**  Six items assessed social closeness to "Anne," from 1 = *strongly disagree* to 5 = *strongly agree* (e.g., "I would feel comfortable being friends with Anne"). The items were adapted from Puhl et al. [49] to refer to "Anne," rather than "the person in this photo." Item scores are averaged, with higher scores reflecting higher levels of social

**Table 2. Participants' scores on the measures at pretest and posttest.**

| Variable | Cronbach's α | Possible Range | Experimental Group (*n* = 49) | Control Group (*n* = 49) |
|---|---|---|---|---|
| | | | *M (SD)* | *M(SD)* |
| Pretest Fat Acceptance | .90 | 1–7 | 5.18 (0.62) | 5.10 (0.66) |
| Posttest Fat Acceptance | .93 | 1–7 | 5.44 (0.61) | 5.23 (0.72) |
| Pretest Attribution Complexity | .84 | 1–7 | 5.88 (0.56) | 5.82 (0.70) |
| Posttest Attribution Complexity | .88 | 1–7 | 6.05 (0.67) | 5.93 (0.71) |
| Pretest Responsibility | .70 | 1–7 | 3.63 (0.68) | 3.67 (0.77) |
| Posttest Responsibility | .78 | 1–7 | 3.94 (0.82) | 3.80 (0.85) |
| Social Closeness to Anne | .84 | 1–5 | 4.52 (0.46) | 4.18 (0.51) |
| Likeability of Anne | .88 | -10–10 | 6.60 (2.94) | 5.96 (2.91) |
| Functionality Appreciation | .86 | 1–5 | 4.39 (0.50) | 4.40 (0.49) |

*Notes.* Cronbach's α was calculated to evaluate the internal reliability of the item scores for the measure of the respective variable.

closeness, or willingness to engage in social activities with "Anne." The scale demonstrated good internal reliability in the original study (α = .92; [49]).

**Likeability of "Anne".** Likeability ratings of "Anne" were made on positivity (-10 = *negative*, 10 = *positive*), agreeableness (-10 = *disagreeable*, 10 = *agreeable*), and sympathy (-10 = *unsympathetic*, 10 = *sympathetic*). The items were adapted from De Ruddere et al. [50] to refer to "Anne," rather than "the patient." Item scores are averaged; higher scores reflect higher likeability of "Anne." De Ruddere et al. did not report internal reliability statistics but the scale demonstrated good construct validity.

**Functionality Appreciation Scale (FAS; [31]).** The FAS comprises 7 items (e.g., "I appreciate my body for what it is capable of doing"), rated from 1 = *strongly disagree* to 5 = *strongly agree*. FAS scores are averaged; higher scores demonstrate higher levels of functionality appreciation. FAS scores have demonstrated good construct validity, test-retest reliability, and internal reliability (α = .86 to.89; [31]).

**Additional items.** To describe the *demographic characteristics* of the sample, participants reported their age, height and weight (to calculate BMI), self-classified weight (1 = *very underweight*, 7 = *very overweight*), ethnic background, and sexual orientation. Note that these data were only used to describe the demographic characteristics of the sample, as summarised in Table 1, and were not planned to be included in the statistical analyses.

Participants also evaluated *"Anne's" perceived body weight* (1 = *very underweight* to 7 = *very overweight*). *Filler items* about imagination were included to align with the cover story (e.g., "As a child, my parents read me story books"); this cover story and subsequent filler items have been used in prior research with functionality-based assignments (cf. [33]). Participants also completed an *awareness check item*, asking them to describe the purpose of the study.

## Statistical analyses

Prior to the main analyses, the data were examined for outliers, defined as any value ±3SD from the group mean, and were adjusted to the boundary value identified (i.e., ±3SD the group mean; [51]). The assumptions with respect to multiple regression analyses were examined [51], and these were all met. To test whether the experimental group reported *lower weight stigma* scores at posttest than the active control group, a series of multiple regression analyses were conducted, one for each weight stigma variable separately. In Block 1, Group (0 = *active control*, 1 = *experimental*), Pretest (where applicable, for the FAAT-based variables), and Functionality Appreciation were entered into the model. Functionality Appreciation was

grand-mean centred (i.e., the sample mean was subtracted from each individual's score on that variable). In Block 2, Group × Functionality Appreciation was added to the model to test whether effects were *moderated* by functionality appreciation. "Lower weight stigma" was operationalised as higher scores on the FAAT scales (i.e., the Fat Acceptance Scale, Attribution Complexity Scale, and Responsibility Scale), and higher scores with respect to social closeness and likeability of "Anne." Functionality appreciation was operationalised using the FAS.

## Results

### Preliminary analyses

There were no missing data at pretest or posttest. Five outliers were identified and adjusted: Three were in the experimental group and concerned scores on Responsibility at pretest ($n = 1$), and on Attribution Complexity ($n = 1$) and likeability of "Anne" at posttest ($n = 1$); Two were in the active control group and concerned scores on Attribution Complexity at pretest ($n = 1$) and on likeability of "Anne" at posttest ($n = 1$). Participants' scores at pretest and posttest are shown in Table 2. As expected given the randomisation of participants to group, there was no significant group difference on any of these scores at pretest, all $ps > .535$, nor did groups differ based on age or BMI, $ps > .309$. Participants' perceived body weight of "Anne" corresponded to "moderately overweight" on the response scale ($M = 6.03$, $SD = 0.68$).

The content of participants' responses to the experimental manipulations was checked. As described above, three participants had not adhered to the instructions, and their data were removed from the dataset. The responses of the remaining participants ($n = 98$) were in line with the instructions of their assigned experimental manipulation. Further, although all participants were required to spend the same amount of time on the writing exercises (15min), the responses of the experimental group included more words ($M = 355.33$, $SD = 155.96$) than the responses of the active control group ($M = 290.35$, $SD = 115.41$), $t(96) = -2.34$, $p = .021$. Note that controlling for word count in the analyses did not change the overall pattern of results, and it was therefore excluded from the models.

With respect to the awareness check, three participants in the experimental group had correctly guessed the aim of the study. However, sensitivity analyses indicated that the removal of their data did not change the overall pattern of results; therefore, their data were retained in the analyses.

### Testing the research questions

For all outcomes, *F* change from Block 1 to Block 2 was nonsignificant, indicating that the addition of Group × Functionality Appreciation did not significantly improve the model (nor were the interaction effects significant). These results show that functionality appreciation *did not moderate* the effects of the intervention approach. Further, in all but one case (Responsibility Scale), Functionality Appreciation did not significantly contribute to the model at Block 1. Therefore, for these outcomes, the model is reported with Functionality Appreciation excluded.

**Fat acceptance.** The overall model significantly predicted fat acceptance at posttest, $F(2, 95) = 185.42$, $p < .001$, adj. $R^2 = .79$. Pretest levels of fat acceptance significantly predicted fat acceptance at posttest, $ß = .88$, $t = 18.95$, $p < .001$. Group also significantly predicted fat acceptance at posttest, $ß = .10$, $t = 2.18$, $p = .032$. In line with the hypothesis, the experimental group demonstrated lower weight stigma, as expressed by *higher levels of fat acceptance*, compared with the active control group.

**Attribution complexity.** The overall model significantly predicted attribution complexity at posttest, $F(2, 95) = 29.46$, $p < .001$, adj. $R^2 = .37$. Pretest levels of attribution complexity

significantly predicted attribution complexity at posttest, ß = .61, *t* = 7.60, *p* < .001. Contrary to hypothesis, there were no group differences in attribution complexity at posttest, as indicated by the nonsignificant effect for Group, ß = .06, *t* = 0.76, *p* = .451.

**Responsibility.** The overall model significantly predicted responsibility at posttest, *F*(3, 94) = 39.64, *p* < .001, adj. $R^2$ = .54. Pretest levels of responsibility significantly predicted responsibility at posttest, ß = .71, *t* = 10.37, *p* < .001, and, in this model only, so did Functionality Appreciation, ß = -.23, *t* = 3.29, *p* = .001. Unexpectedly, there were no group differences in responsibility at posttest; Group, ß = .10, *t* = 1.39, *p* = .167.

**Social closeness to "Anne".** The model significantly predicted social closeness to "Anne," *F*(1, 96) = 11.80, *p* = .001, adj. $R^2$ = .10. Specifically, Group significantly predicted social closeness to "Anne," ß = .33, *t* = 3.44, *p* = .001. In line with the hypothesis, the experimental group demonstrated lower weight stigma, as expressed by *higher levels of social closeness to "Anne"*.

**Likeability of "Anne".** The model did not significantly predict likeability of "Anne," *F*(1, 96) = 1.17, *p* = .28, adj. $R^2$ = .002. Contrary to expectation, there were no group differences with respect to likeability of "Anne;" Group, ß = .11, *t*(97) = 1.08, *p* = .28.

## Discussion

Weight stigma is prevalent across numerous life domains, and negatively affects both psychological and physical health [3,4,9,10,14,15]. Yet, research into weight stigma reduction techniques is limited, and has produced mixed findings [21,22]. Accordingly, this study investigated a novel intervention technique for reducing weight stigma. The primary research question was: Does appreciating the body functionality of another person lead to reductions in weight stigma? As hypothesised, participants who described the body functionality of "Anne," a woman with high BMI, reported lower weight stigma, as expressed by higher levels of fat acceptance and social closeness to "Anne." No group differences were found with respect to attribution complexity, responsibility, and likeability of "Anne." A secondary research question was whether the effects of the approach would be moderated by pre-existing levels of functionality appreciation. The findings did not reveal support for any moderating effect of functionality appreciation.

The present intervention approach was inspired by research from the field of body image, showing that fostering appreciation for one's own body functionality is currently the most effective technique for enhancing positive body image (for a review, see [29]). This study is the first to apply this approach to appreciating another person's body functionality as a technique to reduce weight stigma. In line with objectification theory [39], it was theorised that appreciating another person's body functionality could mitigate weight stigma by counteracting the tendency to evaluate others based on their physical appearance and body weight. Further, appreciating the various domains of body functionality could expand conceptualisations of body functionality to be more complex and holistic—which may be valuable considering that conceptualisations of body functionality of people with high BMI tend to be narrow and negative ("unhealthy," "unfit," "lazy;" [3]). Overall, the present findings support these notions, at least with respect to fat acceptance and social closeness to "Anne." The findings are promising, considering that weight stigma reduction interventions based on attribution theories, which focus on the complex factors that influence body weight, tend to change knowledge about the causes of "overweight" and "obesity," but often do not improve attitudes and acceptance with respect to people with high BMI [21,22]. In fact, it has been proposed that knowledge about the influences on body weight may not be an underlying cause of weight stigma, but rather a means to justify weight stigma [21]. Thus, changes in outcomes such as fat acceptance could potentially be more meaningful and indicative of real reductions in weight stigma.

Interestingly, in the present study we did not find any changes in attribution complexity and responsibility, which capture knowledge about the multiple causes of "overweight" and "obesity" [47].

It is noteworthy that the experimental group experienced higher fat acceptance and positive evaluations towards fat people in general, but also higher social closeness towards "Anne" in particular. These findings are promising because they suggest that the effects of focusing on the body functionality of one person with high BMI could generalise to other people with high BMI, though this should be tested in future research. The findings with respect to social closeness to "Anne" are also valuable considering that positive social interaction plays an important role in reducing negative stereotypes toward perceived out-groups (see *intergroup contact theory*; [52,53]). Some experiments have shown that even just imagining a positive social interaction with a person with high BMI can reduce aspects of weight stigma [54,55].

Given the promising findings with respect to fat acceptance and social closeness, it is unclear why the experimental group did not report higher likeability of "Anne." One possibility is that the active control group also reported liking "Anne." Indeed, when inspecting the mean scores, both groups evaluated "Anne" as rather likeable ($M$ = 5.96 to 6.60 on a scale of -10 to 10). Potentially, writing about "Anne's" home may have led the active control group to imagine her in a more positive and humanising light, beyond any immediate assumptions based on her body weight (cf. objectification theory; [39]), thereby contributing to more favourable evaluations. We did not assess likeability of "Anne," and social closeness, at pretest given the design of our study, whereby "Anne" was only introduced to the participants at the start of the writing exercise, 1 week later. Further, the pretest was timed 1 week prior to the laboratory session, to reduce the risk that participants would guess the aims of the study. Nonetheless, future studies could include an assessment of "Anne" at the start of the laboratory session, after the writing exercise and "Anne" have been introduced, but before the participants have begun writing.

With respect to the absence of group differences in attribution complexity and responsibility, as noted above, knowledge about the multiple influences on body weight may not necessarily cause weight stigma, but may be held as justification for it [21]. Further, of the stigma reduction techniques that aim to enhance knowledge, more explicit techniques are used—for example, whereby participants read about the multiple influences on body weight that are beyond the control of the individual [56]. Investigating weight stigma reduction techniques in isolation is necessary to determine their individual effects. However, multiple complementary techniques will likely be needed to cumulatively change the various expressions of weight stigma.

Last, it is unclear why pre-existing levels of functionality appreciation did not moderate the effects of the intervention approach. The emerging research on functionality appreciation has shown that it is significantly related to other facets of positive body image, including those that incorporate attitudes toward other people's bodies [57]. For example, functionality appreciation is positively correlated with *broad conceptualisation of beauty* [31], which involves perceiving beauty in a variety of body shapes and sizes [58], and is inversely correlated with *appearance-ideal internalisation* [59], which involves endorsing thinness as the "ideal" of beauty [60]. Thus, it was expected that functionality appreciation would impact how an individual responds to describing another person's body functionality. Yet, it could be that appreciation of one's own body functionality does not influence the extent to which one appreciates another person's body functionality. Future research should determine whether the present effects are replicated, and explore other moderators of the effects of stigma-reduction techniques.

## Strengths and limitations

Studies investigating techniques to reduce weight stigma are limited; moreover, the experimental rigour of many of these studies has been questionable [21,22]. The present study is valuable because it investigated a novel, brief, and easy to implement weight stigma reduction technique, using a randomised controlled design, with pretest and posttest assessments of weight stigma. In addition, the present study included assessments of various facets of weight stigma, rather than assessing only knowledge about the influences on body weight or attitudes towards fatness. Nevertheless, this study is not without limitations, which can inform additional directions for future research.

First, the present sample comprised only women. The decision was made to recruit only women to simplify the experimental design, and taking into account that our target person "Anne" was also a woman. Future research could include men, and investigate whether gender moderates intervention effects. Second, although the present sample was diverse in terms of ethnic background, and the proportion of different sexual orientations corresponds to those of the Netherlands more broadly [61], their average age and BMI was relatively young/low, respectively. Future research should include participants who are more diverse in terms of these demographic characteristics. Third, the photograph of "Anne" was of a White woman; though individuals across demographic characteristics experience weight stigma, there is evidence to suggest that demographic characteristics may intersect to determine the extent of experienced weight stigma and its impact [62,63]. Future research could include experimental stimuli with models who vary on other demographic characteristics. Relatedly, future research could include multiple models (also within demographic groups), to enhance the generalisability of intervention effects. Tying back to the importance of including participants of other genders, it will also be valuable to explore how different genders respond to models of their own vs. other genders. Fourth, participants described the body functionality of a person with high BMI, who was perceived as "moderately overweight." It will be valuable to compare the present experimental manipulation to additional groups whereby participants describe the body functionality of a person perceived as "very overweight," and of a person perceived as having lower BMI. Fifth, future research should include follow-up assessments of weight stigma, to determine whether the present effects persist beyond the posttest. Sixth, we cannot rule out the possibility that participants responded to the study measures in a socially desirable manner. Importantly, however, scores on the FAAT have not been correlated with socially desirable responding ([47]; similar information for the items assessing social closeness and likeability is not available). Last, some of the study measures assessed facets related to behaviour (e.g., self-reported desire to engage in social activities with "Anne"), but future research should also include direct assessments of behaviour.

## Conclusions

This study investigated a novel technique to reduce weight stigma, whereby participants reflected on the body functionality of a person with high BMI. Encouragingly, the experimental group reported higher levels of fat acceptance and social closeness to "Anne." However, future research is needed to examine whether these effects can be replicated, as this is the first study to test this technique. We believe that the onus for stigma-reduction should fall on those who stigmatise others, rather than on the individuals who experience stigma, or on solely teaching them techniques to cope with stigma. In this light, future research could explore how to amplify the reach of weight stigma reduction strategies. For example, should fostering appreciation for others' body functionality reliably mitigate weight stigma, it will be interesting to investigate how media portrayals (which typically perpetuate weight stigma; [64]) can

emphasise functionality appreciation in others. Emerging research on body-positive media, including portrayals of women of diverse body sizes engaging in joyful physical activity [65], have been shown to enhance positive body image (for a review, see [66]). It will be valuable to test whether these media portrayals also reduce weight stigma by fostering functionality appreciation for others. We hope that this study serves as an inspiration for future endeavours in this understudied yet important area of research.

## Supporting information

**S1 File. Experimental manipulations: Writing exercise instructions.**
(DOCX)

**S2 File. Photograph of "Anne".**
(DOCX)

## Author Contributions

**Conceptualization:** Jessica M. Alleva, Kai Karos, Angela Meadows, Moon I. Waldén, Francesca Lissandrello, Melissa J. Atkinson.

**Formal analysis:** Jessica M. Alleva.

**Investigation:** Moon I. Waldén, Francesca Lissandrello.

**Methodology:** Jessica M. Alleva, Kai Karos, Angela Meadows, Moon I. Waldén, Francesca Lissandrello, Melissa J. Atkinson.

**Supervision:** Jessica M. Alleva.

**Writing – original draft:** Jessica M. Alleva.

**Writing – review & editing:** Jessica M. Alleva, Kai Karos, Angela Meadows, Moon I. Waldén, Sarah E. Stutterheim, Francesca Lissandrello, Melissa J. Atkinson.

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
