## [Decision Letter · Decision Letter 0]

24 Mar 2021

PONE-D-21-01302

“What can her body do?” Reducing weight stigma by appreciating another person’s body functionality

PLOS ONE

Dear Dr. Alleva,

Thank you for submitting your manuscript to PLOS ONE. After careful consideration, we feel that it has merit but does not fully meet PLOS ONE’s publication criteria as it currently stands. Therefore, we invite you to submit a revised version of the manuscript that addresses the points raised during the review process.

We look forward to receiving your revised manuscript.

Kind regards,

W. Douglas Evans, PhD

Academic Editor

PLOS ONE

Journal Requirements:

Additional Editor Comments (if provided):

Please respond to all reviewer comments and resubmit a revised manuscript.

Reviewers' comments:

Reviewer's Responses to Questions

**Comments to the Author**

1. Is the manuscript technically sound, and do the data support the conclusions?

Reviewer #1: Yes

Reviewer #2: Yes

2. Has the statistical analysis been performed appropriately and rigorously? 

Reviewer #1: Yes

Reviewer #2: Yes

3. Have the authors made all data underlying the findings in their manuscript fully available?

Reviewer #1: No

Reviewer #2: Yes

4. Is the manuscript presented in an intelligible fashion and written in standard English?

Reviewer #1: Yes

Reviewer #2: Yes

5. Review Comments to the Author

Reviewer #1: The manuscript examined an intervention to reduce weight stigma. This was a simple experimental design with two conditions: The intervention, which asked participants to write about the functionality of “Anne’s” body, and the control condition, which asked participants to write about “Anne’s” house. The research was well designed. Pre-test levels of variables were taken a week before the intervention to reduce demand characteristics. The paper is clear, concise, and well-written.

I just had a few minor issues:

1. In the introduction, what is described as “other-objectification” is just plain objectification. No need to add the “other-“

2. Analyses: The authors should include word count as a covariate in all analyses.

3. Analyses: It was unclear to me whether the authors had planned to include demographic covariates in any analyses. Please clarify rationale.

4. Analyses: Why take out Functionality Appreciation as a covariate when that was in the preregistered plan?

5. Discussion: Nice job with the limitations. However, there is no need to italicize the names of the scales or other words you want to highlight. The writing is clear and enough for emphasis.

I really enjoyed reading this paper!

Reviewer #2: This paper reports on a weight-stigma reduction technique, which in and of itself is important as very little work has investigated reduction of this prevalent stigma. Even better, this paper reports evidence that this technique seems to work. I have the following comments to help present this work in the best possible light.

Introduction

• Because there are now a good number of review papers evincing the negative consequences of weight stigma, I’d recommend condensing this introduction down to summarize the findings from the reviews rather than picking and choosing primary sources.

• What could use some more unpacking are the reviews cited on page 4, line 67. Please describe the nature of these mixed results, which types of interventions seem to work or not work, why this might be the case. This will give a better setup to then discuss the aim of the present study.

• The paragraph beginning on Page 4 line 86 might make more sense if flipped with the previous paragraph – at least the pieces evincing the efficacy of functionality-based approaches. That is, the theory should be introduced before the research supporting it.

Method

• The method is rigorous and described and justified well.

Discussion

• Expanding upon the gender limitation, it would also be helpful for future research to examine gross-gender bias (i.e., female participants rating a male target; male participants rating Anne).

• Another limitation that should be measured is that these findings get at only the bias side of stigma. Future research should look into behavioral outcomes that might get at propensity for discrimination.

• The discussion does a nice job of outlining limitations in terms of future research that can come from them.

• Only one other comment: while I agree that this work may have implications for society/media etc. It’s important that the authors hedge their conclusions to not over-sell a rather preliminary study of 100 participants.

6. PLOS authors have the option to publish the peer review history of their article (what does this mean?). If published, this will include your full peer review and any attached files.

Reviewer #1: No

Reviewer #2: No

---

## [Author Response · Author response to Decision Letter 0]

7 Apr 2021

Response to Reviewers

We would like to thank the Academic Editor and the two Reviewers for the time and effort they have invested in our manuscript, and for their helpful suggestions for improvement. Below, we outline how we have responded to each comment. In the marked manuscript, the changes are indicated in blue font. 

Academic Editor

Response: Thank you. We have now ensured that our manuscript meets the Journal’s style requirements for the manuscript formatting and the file naming. In line with these requirements, the referencing style has been changed from APA to Vancouver. 

Response: Thank you. We will indeed make the data available in a repository at acceptance (if applicable), and we understand that the manuscript would be held until we provide the relevant accession numbers or DOIS necessary to access our data. 

Response: Thank you. We have now included captions for our Supporting Information files at the end of our manuscript, and have updated in-text citations accordingly. 

Response: Thank you. We have reviewed our reference list and ensured that it is complete and correct. We have not cited any papers that have been retracted. Note that we have replaced one of our references (Alberga et al., 2016) with another reference (Lee et al., 2014), which we believed to be more suitable to this manuscript. Alberga et al. (2016) focused on weight-stigma reduction techniques among health care professionals specifically. Lee et al. (2014) focused on techniques among the broader population, and report effect sizes by intervention approach.

Additional Editor Comments (if provided):

Please respond to all reviewer comments and resubmit a revised manuscript.

Response: Thank you. We have responded to all reviewer comments below, and our revised manuscript is enclosed (both a marked and a “clean” version). 

Reviewer #1

The manuscript examined an intervention to reduce weight stigma. This was a simple experimental design with two conditions: The intervention, which asked participants to write about the functionality of “Anne’s” body, and the control condition, which asked participants to write about “Anne’s” house. The research was well designed. Pre-test levels of variables were taken a week before the intervention to reduce demand characteristics. The paper is clear, concise, and well-written.

Response: Thank you for your compliments about our study design and the manuscript. We appreciate your helpful suggestions for further strengthening our manuscript.

I just had a few minor issues:

1. In the introduction, what is described as “other-objectification” is just plain objectification. No need to add the “other-“

Response: Thank you, good point! We have now removed the word “other” (p. 5-6).

2. Analyses: The authors should include word count as a covariate in all analyses.

Response: Thank you for this suggestion. Respectfully, we are hesitant to include word count as a covariate in the analyses reported in the manuscript. All participants were required to spend the same amount of time on the writing exercise (15 minutes). We do not believe that word count is a good indicator of participant engagement with the exercise. For example, some participants may have taken more time to think about their answers, before writing them down, or some participants may have been faster at typing. Importantly, participants were in a room that only contained the computer and the writing exercise materials. They did not have access to their phones or other potential distractions, and so we are confident that participants indeed spent the 15 minutes on their writing exercises, regardless of the number of words written. The experimenter waited outside the laboratory, and kept track of the time, only re-entering to open the remaining questionnaires after the 15 minutes had passed. 

Nevertheless, as a check, we had re-run all of the analyses including word count as a covariate. We confirm that the inclusion of word count as a covariate did not change the overall pattern of results. We have made note of this in the manuscript (p. 15), but retained our original analyses for clarity and given our reasoning above, about the meaningfulness of the word count data. We also believe that the exclusion of word count from the models, taking into account that they do not change the overall pattern of results, is in the best interest of the parsimony and interpretability of the statistical models. 

3. Analyses: It was unclear to me whether the authors had planned to include demographic covariates in any analyses. Please clarify rationale.

Response: We clarify that we did not plan to include any demographic information in any analyses (cf. our pre-registration on AsPredicted). The demographic information was only collected as a means to characterise the sample, and is reported in Table 1 for this purpose. We have now clarified this in the manuscript as well (p. 13). 

4. Analyses: Why take out Functionality Appreciation as a covariate when that was in the preregistered plan?

Response: In the preregistered plan, we described that, “To test our secondary hypothesis, concerning functionality appreciation as a moderator of intervention effects, we will conduct a series of regression analyses, with Group as predictor, and Functionality Appreciation as moderator. For the analyses concerning the FAAT, Pretest will also be included as a predictor.” 

We had not specified the more detailed steps of these analyses. However, we had indeed planned to exclude Functionality Appreciation as a covariate when it did not significantly contribute to the model. This decision was made early on in the study design process, prior to preregistration, but we regret that we did not describe these finer details in AsPredicted. We would be happy to provide our time-stamped correspondence with our department’s statistician, as support for the integrity of this decision. 

Note that the exclusion of non-significant predictors from a statistical model is a valid decision. In our case, functionality appreciation did not contribute meaningfully to the models (with the exception of the outcome responsibility, where it was retained). The exclusion of superfluous predictors from statistical models increases the parsimony of the models, as well as the interpretability for the reader. 

5. Discussion: Nice job with the limitations. However, there is no need to italicize the names of the scales or other words you want to highlight. The writing is clear and enough for emphasis.

Response: Thank you for your kind words about our Limitations section. We have now removed italics where necessary. We have only retained italics when new terms or names of theories were introduced. This is typical in our field, but we are happy to remove these italics as well, if desired. 

I really enjoyed reading this paper!

Response: Thank you for your enthusiasm about our manuscript. Again, we really appreciate your helpful suggestions for further improvement.

Reviewer #2

This paper reports on a weight-stigma reduction technique, which in and of itself is important as very little work has investigated reduction of this prevalent stigma. Even better, this paper reports evidence that this technique seems to work. I have the following comments to help present this work in the best possible light.

Response: Thank you for your kind words about our manuscript and the present intervention technique. We are grateful for your suggestions for improving our manuscript.

Introduction

• Because there are now a good number of review papers evincing the negative consequences of weight stigma, I’d recommend condensing this introduction down to summarize the findings from the reviews rather than picking and choosing primary sources.

Response: Thank you for this suggestion. Respectfully, we have decided to retain the more detailed description of the negative consequences of weight stigma. Though these negative consequences are known among weight-stigma scholars, our experience has taught us that most outside the field are not aware of these negative consequences, or the extent of them. As we hope to eventually have our manuscript read by people both within and outside the field of (weight) stigma, and beyond academia, we think it is important to retain these details as they help to underscore the importance of this topic, and to provide readers with concrete examples of how weight stigma is harmful. Note that with the change from APA to Vancouver referencing styles, this paragraph has been condensed slightly in length. 

• What could use some more unpacking are the reviews cited on page 4, line 67. Please describe the nature of these mixed results, which types of interventions seem to work or not work, why this might be the case. This will give a better setup to then discuss the aim of the present study.

Response: Thank you for this suggestion. We have now written a paragraph that unpacks the reviews we have referenced (note that we have replaced Alberga et al., 2016, with Lee et al., 2014; see our comment in response to the Academic Editor above). We believe this has indeed strengthened the set up for the aim of the present study – thank you! (p. 3-4)

• The paragraph beginning on Page 4 line 86 might make more sense if flipped with the previous paragraph – at least the pieces evincing the efficacy of functionality-based approaches. That is, the theory should be introduced before the research supporting it.

Response: Thank you for this suggestion. We understand that the storyline might be unconventional, beginning first with the description of the intervention approach before describing its underlying theory. Respectfully, we have retained this order of the storyline, because we believe it provides a better flow for the Introduction, whereby we now unpack the prior reviews and intervention approaches, before describing the present intervention approach, and its scientific support. Experimenting with reversing the order of these paragraphs led to a less strong storyline, in our view.

Method

• The method is rigorous and described and justified well.

Response: Thank you for your kind words about the Method of our study.

Discussion

• Expanding upon the gender limitation, it would also be helpful for future research to examine gross-gender bias (i.e., female participants rating a male target; male participants rating Anne).

Response: Thank you, this is an interesting avenue for future research! We have now included this in the Limitations section (p. 21).

• Another limitation that should be measured is that these findings get at only the bias side of stigma. Future research should look into behavioral outcomes that might get at propensity for discrimination.

Response: Thank you, we agree with this suggestion! We had already included this in our Limitations section (see the final sentence of the Limitations section on p. 22). However, if you would like us to elaborate on this point, or rephrase this sentence, please let us know.

• The discussion does a nice job of outlining limitations in terms of future research that can come from them.

Response: Thank you. We are pleased to hear that the Limitations point toward directions for future research.

• Only one other comment: while I agree that this work may have implications for society/media etc. It’s important that the authors hedge their conclusions to not over-sell a rather preliminary study of 100 participants.

Response: Thank you for this suggestion. We have now modified the concluding sentence on p. 22 to underscore that this is the first study on this technique, and thus future research is needed to examine whether these results can be replicated.

---

## [Editor Report · Decision Letter 1]

28 Apr 2021

“What can her body do?” Reducing weight stigma by appreciating another person’s body functionality

PONE-D-21-01302R1

Dear Dr. Alleva,

We’re pleased to inform you that your manuscript has been judged scientifically suitable for publication and will be formally accepted for publication once it meets all outstanding technical requirements.

Kind regards,

W. Douglas Evans, PhD

Academic Editor

PLOS ONE
---

## [Editor Report · Acceptance letter]

4 May 2021

PONE-D-21-01302R1 

“What can her body do?” Reducing weight stigma by appreciating another person’s body functionality 

Dear Dr. Alleva:

I'm pleased to inform you that your manuscript has been deemed suitable for publication in PLOS ONE. Congratulations! Your manuscript is now with our production department. 

Kind regards, 

on behalf of

Professor W. Douglas Evans 

Academic Editor

PLOS ONE